# LGBTQ+ Psychosocial Concerns in Nursing and Midwifery Education Programmes: Qualitative Findings from a Mixed-Methods Study

**DOI:** 10.3390/ijerph182111366

**Published:** 2021-10-29

**Authors:** Michael Brown, Edward McCann, Gráinne Donohue, Caroline Hollins Martin, Freda McCormick

**Affiliations:** 1School of Nursing and Midwifery, Queen’s University Belfast, Belfast BT97BL, UK; freda.mccormick@qub.ac.uk; 2School of Nursing and Midwifery, Trinity College Dublin, D02 T283 Dublin, Ireland; mccanned@tcd.ie (E.M.); donohuga@tcd.ie (G.D.); 3School of Health & Social Care, Edinburgh Napier University, Edinburgh EH11 4BN, UK; c.hollinsmartin@napier.ac.uk

**Keywords:** LGBTQ, mental health, psychosocial, education, nursing, midwifery

## Abstract

LGBTQ+ people experience significant physical and psychosocial health issues and concerns, and encounter barriers when accessing healthcare services. We conducted a mixed-methods research study across all Schools of Nursing and Midwifery in the United Kingdom and Ireland using a survey and qualitative interviews. This was to identify the current content within nursing and midwifery pre-registration programmes in relation to LGBTQ+ health and to identity best practice and education innovation within these programmes. The survey was completed by 29 academics, with 12 selected to participate in a follow-up in-depth qualitative interview. Analysis of the data from the survey and interviews identified five themes: there is variable programme content; academics are developing their own programmes with no clear consistency; LGBTQ+ health is being linked to equality and diversity; there are barriers to education provision; and these is some evidence of best practice examples. The findings of the study support the need to develop and implement a curriculum for LGBTQ+ health in nursing and midwifery pre-registration programmes with learning aims and outcomes. Academics need support and tools to prepare and deliver LGBTQ+ health content to nurses and midwives as they ultimately have the potential to improve the experiences of LGBTQ+ people when accessing healthcare.

## 1. Introduction

The term LGBTQ+ will be used throughout this paper as it is widely accepted and encompasses all sexual and gender identities and groups [1]. The provision of accessible and appropriate healthcare for all citizens is a global ambition, yet for many communities remains elusive and aspirational [2,3]. Despite these aspirations, health inequalities and significant barriers continue to be a reality for many populations and communities, and impact negatively on their general health and well-being, including biopsychosocial issues and concerns [4,5,6,7,8].

There has been growing attention to the specific health inequalities experienced by many LGBTQ+ people and the barriers that may exist when seeking access to appropriate care and support [9]. Some LGBTQ+ people and their families have experienced historical discrimination and stigma when accessing and using care and support services, including health care [10,11]. A significant number of LGBTQ+ people have reported the negative impact of experiencing minority stress including homophobia, transphobia and biphobia [12]. Research evidence highlights the often poorer physical and mental health experienced across LGBTQ+ sub-populations [13]. From a physical health perspective, there is increased cardiovascular and respiratory disorders and higher incidences of sexual health conditions such as sexually transmitted infections (STIs) [14,15]. In terms of mental health, many conditions are over-represented, such as depression, suicidality, substance use, anxiety and trauma [16,17]. Further, many LGBTQ+ people have specific cultural and psychosocial needs that require specific consideration and responses from healthcare providers [18].

It is estimated that globally there are some 59 million health care workers such as doctors, nurses, midwives and other allied health professionals [19]. It is recognised that healthcare workers, including nurses and midwives, are a valuable and important resource vital to improving the health and well-being of individuals, families, populations and communities [20]. Of the global population of health professionals, nurses and midwives comprise the largest single group, with the current figure standing at some 28 million. Furthermore, they comprise 50% of the healthcare workforce and play important roles in reducing health inequalities and providing direct care and support to patients, families and populations [21]. In the United Kingdom (UK) and Ireland, there are approximately 750,000 nurses and midwives registered with the Nursing and Midwifery Council (NMC) in the UK [22] and with the Nursing and Midwifery Board for Ireland (NMBI) [23]. From the perspective of the needs of LGBTQ+ people, their families and their communities, health professionals play important roles that should be underpinned by relevant education and ongoing professional development [24,25].

Therefore, to effectively address the health concerns of LGBTQ+ people, nursing and midwifery practice must be based upon recognising and responding to the cultural differences of individuals, families, communities and populations on a global basis. To this end, nursing and midwifery practice needs to be informed by Transcultural Theory, with knowledge and understanding of the cultural needs of populations. This includes recognising and responding to transcultural differences, and their specific and distinct care and support needs and the responses necessary from health services [26]. To achieve this, health professionals need to be prepared with the necessary knowledge and skills to address the cultural and distinct health needs of LGBTQ+ people, their families and communities. Where nurses and midwives are concerned, as the largest group of health professionals, they are in an influential position to provide quality transculturally competent healthcare specific to the needs of LGBTQ+ people, their families and communities. However, despite the apparent need for culturally competent care and support, education provision focusing on LGBTQ+ health concerns within nursing and midwifery in the UK and Ireland appears to be limited and is an area requiring further investigation [24,25]. Therefore, it is necessary to more fully understand the scope and extent of current education provision delivered for nurses and midwives as part of their pre-registration education programmes. Hence, the aim of this study was to identify how nursing and midwifery education programmes address the psychosocial concerns of LGBTQ+ people across the UK and Ireland.

## 2. Materials and Methods

The aim of the study was to identify how nursing and midwifery education programmes across the UK and Ireland address the psychosocial concerns of LGBTQ+ people.

The study questions were:What content is delivered within nursing and midwifery pre-registration programmes to address the psychosocial concerns of LGBTQ+ people?What are the barriers to the inclusion of LGBTQ+ health content within nursing and midwifery pre-registration programmes?What are the areas of best education practice regarding the inclusion of LGBTQ+ health concerns in nursing and midwifery pre-registration programmes?

### 2.1. Study Design

This study utilised a mixed-methods approach which involved an anonymous online questionnaire and qualitative interviews. Information about the study was provided to all respondents before commencing the questionnaire. An opt-in option was provided to questionnaire respondents if they agreed to be contacted by the researchers for consideration to participate in a follow-up qualitative interview. In advance of the individual interviews, all participants received a participation information leaflet and signed and returned a completed consent form. This approach enabled the identification of some of the LGBTQ+ health activity taking place in nursing and midwifery programmes across the UK and Ireland, and to explore current programme content and best practice and education innovation through in-depth interviews with nursing and midwifery academics working in Schools of Nursing and Midwifery.

### 2.2. Ethics

Prior to commencing the study, Research Ethics Committee approval was granted with all ethical processes followed throughout.

### 2.3. Study Participants

Information about the study was circulated by email to all Heads of Schools of Nursing and Midwifery in the UK and Ireland (*n* = 135). The Head of School was asked to forward the information about the study to an administrator. To avoid perceived coercion, the administrator identified academics within the school responsible for the delivery of LGBTQ+ content in the nursing and midwifery programmes and forwarded the invitation to participate in the online survey. This yielded a total of *n* = 29 responses (39%) from October 2020 to March 2021. Consent to be contacted to opt-in to participate in a follow-up individual qualitative interview was provided by 21 respondents, from which a purposive sample of nursing and midwifery academics was identified, with 12 participants selected. Respondents who took part in the questionnaire and those who participated in an interview represented both nursing and midwifery. Table 1 provides participant information by programme. Consideration was also given to asking respondents their sexual orientation, age, and nationality. However, as individuals were responding on behalf of their institution, this was considered unnecessary and intrusive. Study participants were recruited from Schools of Nursing and Midwifery in England, Scotland, Wales, Northern Ireland and the Republic of Ireland. The responses by country are presented in Figure 1.

## 3. Data Collection

An online anonymous questionnaire comprising a total of 36 items using Microsoft Forms was used to collect background data. The questionnaire included demographic details regarding the nursing and midwifery programmes and the focus on LGBTQ+ psychosocial needs and concerns within nursing and midwifery pre-registration programmes. Responses were required in the form of choosing from five options—”non-existent”, “limited”, “adequate”, “moderately adequate” and “fully adequate”. Preferences were rated in order of importance supported by a free text option for further details. Individual semi-structured interviews were held via Microsoft Teams lasting from 30 to 53 min. Qualitative interviews were conducted using a semi-structured interview guide developed and piloted by the research team, comprising open-ended questions to prompt participants to provide detailed responses regarding current curriculum activity and best education practice in relation to addressing LGBTQ+ psychosocial needs and concerns content in nursing and midwifery pre-registration programmes. Follow-up questions were used as part of the process to obtain clarity and depth of data. All interviews were recorded and transcribed verbatim. Anonymity of each transcript was assured by removing all identifiable information and assigning each participant a gender-neutral pseudonym.

### Data Analysis

Central to qualitative data analysis is the application and maintenance of comprehensive methods of data collection, analysis and synthesis. These approaches were rigorously followed by the researchers to ensure the credibility, trustworthiness and dependability of the data analysis process [27,28]. The free text responses in the questionnaire were extracted and analysed along with the individual interviews to form the qualitative dataset and enable the analysis process [29]. Each member of the research team (M.B., E.M. and F.M.) independently read the transcripts to gain an understanding of the content and identify data relevant to the aim and questions of the study. Then the research team collectively analysed the data into sub-themes, and cross-checked and agreed the main themes and sub-themes [30]. As a result of this robust process, how nursing and midwifery education programmes address the psychosocial concerns of LGBTQ+ people and areas of best education practice were identified, thereby enabling the potential transferability of the study findings more widely [31].

## 4. Results

Five broad themes emerged from the data analysis and synthesis: (i) variable programme content; (ii) developing programme consistency; (iii) linking LGBTQ+ health to equality and diversity; (iv) barriers to education provision; and (v) best practice examples.

### 4.1. Variable Programme Content 

There was a willingness by academics to embrace LGBTQ+ health and psychosocial needs and concerns, and the extent to which this was included and embedded in programmes was described as a “seed, yet there remains a gap, a massive gap”. Although there was concern that there is a lot of “lip service” towards the LGBTQ+ community, some academics were taking the initiative and driving related education forward by seeking ways and opportunities to integrate the issues within nursing and midwifery programmes.

“As well as a passion for LGBTQIA health, the concepts of social justice and intersectionality are very at the forefront of my thinking. When we started developing the programmes, I actually put forward the idea that we had… a module within adult nursing that specifically looked at social justice, using intersectionality and focused on the protected characteristics from the Equalities Act.”(Ryan)

“I personalised it and I deliberately personalised it so that they could really appreciate quite how alienated that would be and then from there people understanding the damage of heteronormative language and the forms and the attitudes… Obviously to be constructively aligned within the programme I have to do some sessions about the impact on others… I talk about diversity and their pre-existing mental health. One of the areas that I explore is LGBT+ communities and also LGBT+ midwives.”(Charlie)

However, although some academics were developing LGBTQ+ programme content of their own volition, there was a need for clear guidance and direction to help ensure that they “got it right”. There was a clear desire for LGBTQ+ health to be incorporated in the nursing and midwifery pre-registration curriculum to ensure that all students were consistently informed and equipped to respond effectively to the wide and diverse LGBTQ+ psychosocial issues that they would inevitably encounter in their future practice as registered nurses and midwives.

“I am 59 and I am a gay man but for somebody who is 59 and for a woman who’s 19, who’s a lesbian, my experience of being gay is very different from hers, or could be, or it could be somebody in their 70′s”.(Alex)

“Because it is not required to be built into the programme, it doesn’t exist. Again, something that is policy driven and the nursing and midwifery regulators could ensure through their monitoring and programme review processes is visible and included.”(Lee)

### 4.2. Developing Programme Consistency

The inclusion of LGBTQ+ health where it does exist within nursing and midwifery programmes takes many forms. There were examples of formal lectures and workshops, in addition to examples of online learning and input from LGBTQ+ service users “to give their stories”.

“Usually it’s classroom, lectures, blended learning, some online things. All of them, service users were key. The mental health person, the nursing and the midwives were using service users to give their stories, and I have just told you I have done the same. Stories and service users are very key.”(Sam)

“We start with the concept and then we build towards the nursing practice implications. We find that really helps them embed it and really understand it…With the mental health study days that I do, again I turn that into talking to them about how do you support trans youth; how do you look for signs of gender dysphoria as opposed to somebody who is trans. Not every trans person has gender dysphoria. This idea of labelling everybody as dysphoric when that’s not necessarily everybody’s story.”(Ryan)

However there appeared to be no clear consistency as to what was included within programmes and a lack of a rationale to support a structured and systematic education focus on the needs of LGBTQ+ people across the curriculum. However, some nursing and midwifery programmes had been developed to reflect the changing society and psychosocial needs of patients, families and communities, including LGBTQ+ people. Other programmes, in contrast, lacked a clearer focus, with content being left to individual academics, with some feeling poorly prepared and lacking in confidence. 

“I think we have got a very clear inclusion of trans in the curriculum now. I think that again is how society has shifted agendas.”(Finn)

“I was approached by the local perinatal mental health team… they brought up about training around LGBTQ because they said that we’ve got more people who are trans becoming pregnant and they are not having good experiences of maternity care, so we need to do some education on that. We had inclusion around LGBTQ issues, but we hadn’t included about trans people… we’re getting trans people become pregnant, but the staff aren’t prepared to support them. I know from talking with the LGBTQ forum they said that. They know that some trans people are reluctant to attend for ante-natal care because of the reaction they get which is upsetting for me really.”(Blake)

There was recognition, notably in midwifery services, that healthcare professionals were encountering more diverse situations, including interactions with the LGBTQ+ communities, yet there was a perceived lack of knowledge and competence in how to respond.

“I still think midwifery academics have got a way to go. Some of the language that we use makes assumptions about sexuality and this needs to change.”(Charlie)

“I think we have to be more open, be more open, and not assuming. And I’d say, really understand the psychosocial issues much further, particularly those who I find teach labour ward skills are still very medically focused. I think the whole concept of the psychosocial issues we need to address much more than we are doing.”(Blake)

### 4.3. Linking LGBTQ+ Health to Equality and Diversity

Although there were examples of attempts to integrate and reflect LGBTQ+ health within nursing and midwifery programmes, these were often linked to generic equality and diversity training covering general topics such as anti-discriminatory practice, communication, and language, as opposed to focusing specifically on the LGBTQ+ health needs and wider psychosocial concerns and the nursing and midwifery responses required. There were no examples of, for example, standalone healthcare modules with a LGBTQ+ focus.

“It all comes under the umbrella of diversity. They are able to care for women with diverse needs. How diverse that is depends. We have got cultural-specific stuff, but it’s broad. Nothing LGBT specific.”(Blake)

“I spoke with our Equality and Diversity Student Services Lead, and we discussed how we could ensure that LGBT is more integrated into the curriculum… I said that I felt that it was really important it was embedded rather than an add on… the Programme Director makes sure that we are all mapped fully against the NMC Standards, but also against our equality plan.”(Finn)

Although the links to the equality and diversity agenda were welcome and provide a wider framework, there was recognition that this approach alone would not meet the needs of nursing and midwifery students to equip them with the skills and knowledge required. A clearer specific focus and thread throughout and across the nursing and midwifery programmes was viewed as necessary to enable students to respond effectively to the psychosocial needs of gay men, lesbians, bisexuals and trans people within the context of LGBTQ+ health issues they will encounter in practice.

“I think it is just really important that students see themselves reflected within the curriculum. And students see the patients that they are going to meet reflected within the curriculum. If your curriculum is traditional, it is whitewashed, everybody is heterosexual from a middle-class background, that kind of thing, it is not reflective of real life and it’s not reflective of your student population either.”(Chris)

“One of the issues coming up for trans people at the moment is for someone that’s transitioned earlier in life, and now if they have got dementia, it may mean that they don’t recognise their genitals or they wonder why they are being called by a name that wasn’t the name assigned at birth, that type of thing.”(Joe)

### 4.4. Barriers to Education Provision

Three main barriers were identified regarding the inclusion of LGBTQ+ health content within nursing and midwifery pre-registration programmes: (i) personal confidence; (ii) limited knowledge and skills of LGBTQ+ issues and needs; and (iii) identification and availability of networks.

#### 4.4.1. Personal Confidence

The confidence of academics in delivering LGBTQ+ health related issues varied, with some questioning their suitability and feeling vulnerable and lacking in confidence. Nevertheless, and despite their personal concerns and perceived limitations, they endeavoured to embrace the topic and engage with students. In other cases, the academic was gay and had taken on the role of ensuring that LGBTQ+ health was visible in the nursing and midwifery programme.

“In many ways I don’t feel I was the right one. I got the job by being willing rather than having any great experience in the area… Sometimes I will say to the students, I’m not an expert in this field. I still to now struggle with the use of pronouns… Some of that may be the age I grew up in… Yes, that’s it, I’m not sure I’m necessarily the person to take it on because how well do I understand the issues… I don’t feel I’m the best person to teach here. I just appeared to be the only person who was willing to teach it at the beginning.”(Ray)

“I actually probably think it would be better coming from someone who is a member of the LGBTQ+ community. I feel a bit of a fraud trying to advocate for them… I think it can be something that some staff feel that they are not fully well equipped to deliver and can feel sometimes a little bit uncomfortable about.”(Chris)

#### 4.4.2. Limited Knowledge and Skills of LGBTQ+ Issues and Needs

The LGBTQ+ knowledge and skills acquired by many academics had emanated mainly from experiences they had encountered either personally or through listening to family and friends. They relied on this knowledge to inform their teaching and address psychosocial issues when delivering LGBTQ+ health in the classroom. Many recognised that this was not ideal, appreciating the need for further development and support.

“I have brought my own experiences as a gay man, but I am very mindful that I am a gay man and not a lesbian or bisexual and I am still learning a lot, but I suppose I have my own insights.”(Alex)

“She [daughter] had gone into hospital and straightaway there was an assumption made that she was very heterosexual and when her partner turned up, they couldn’t bring themselves to use the word partner… Because of that and understanding quite how challenging that was to her mental health at a time where she was particularly vulnerable, that sort of informed my philosophy around how I was going to teach that.”(Charlie)

Academics were also responsible for researching and preparing their teaching and learning resources and materials, for which they drew upon a range of sources. It was important for academics to capture the psychosocial issues, and conversations with the LGBTQ+ community, and their family and friends often informed the content of their sessions with the students.

“We have a set of resources from service users and local members of the community who are happy to share their story and work with students. Literally at the moment I have got the YouTube video popped up and open… A gentleman who is now in his fifties who talks about his transition from female to male in his twenties and talks about the challenges with healthcare. I am reviewing that now and looking to get that integrated where I can.”(Lee)

“When I go down and see my daughter… I often meet a lot of her friends…They are often quite negative, her friends, because they have experienced bad situations. I can use that as a vehicle to address in my lectures. Sometimes I’ll say to them, you are making a really good point, do you mind if I talk about this conversation with my students. They always say yes. They always say yes, go ahead, fill your boots, I want this message to be heard.”(Charlie)

#### 4.4.3. Identification and Availability of Networks

The need to involve the LGBTQ+ community was recognised and emphasised by many of the respondents. The first-hand experiences and insights that members of the LGBTQ+ communities could bring to the education of nursing and midwifery students were valued by the academics. There was recognition by some respondents that there were people in other departments across the university who would also be well placed to contribute to the nursing and midwifery programmes, such as academic colleagues in social work and education faculties. However, identifying relevant groups and engaging willing participants from local LGBTQ+ organisations was problematic in some locations.

“I always love it if sessions like this could be delivered by people who identified with the group that they were talking about. But I do recognise actually we can’t always do that and as long as we have materials that are developed in a person-centred way in a way that reflects language that is positive and all that sort of stuff, I think we can work within it. But it’s not ideal.”(Ryan)

“It’s really important to get LGBTQ input into this wherever possible, but the challenge is how do you go about sourcing that when it’s still a sensitive issue for some.”(Lee)

### 4.5. Best Practice Examples

The areas of best education practice that enabled the inclusion of LGBTQ+ health concerns in nursing and midwifery pre-registration programmes can be linked to several areas of pedagogical activity: skills simulation, linking in with local networks and LGBTQ+ champions and success.

#### 4.5.1. Skills Simulation

The qualitative data from the questionnaire revealed that skills simulation appeared to be rarely included in an LGBTQ+ context. However, through the detailed analysis of the qualitative interviews, participants provided examples of the use of skills simulation in areas such as developing student knowledge and skills regarding the needs of transgender people, same sex parenting and surrogacy.

“… role play in relation to having somebody who is playing the part of somebody from the LGBT community whether that is the midwife, the birth partner, or the woman in the bed, and just to play it out in that respect.”(Jordon)

“We try to incorporate in a realistic way rather than throwing everything at them. As they work their way through a skills scenario for example in our skills lab, they will move from patient to patient and within that there may be a same sex parent, and then the other side, maybe a hetero-sexual person of colour with perhaps HIV. We integrate it.”(Finn)

The benefit of incorporating an LGBTQ+ context within skills simulation was also regarded as valuable by some participants who used it in other programmes or had not considered doing so within their current nursing and midwifery pre-registration programmes. Skills simulation with a LGBTQ+ focus was seen as an area of new untapped potential.

“It was really seeing the ideas brought to life but in a safe environment, in a simulated environment, as opposed to being in practice meeting somebody and then putting their foot in it.”(Ryan)

“You could definitely use simulation to really explore in a safe environment what it means because I think a big part of it for both staff and students is they don’t want to offend. But by trying to avoid offending they remain silent so therefore people remain invisible. It’s how to think actually you are better off to be more sensitive and ask the questions in an inclusive manner. If you make some mistakes people will forgive you rather than not saying anything and therefore, we don’t see, we don’t hear, we don’t learn. Simulation would be perfect, absolutely perfect.”(Alex)

#### 4.5.2. Linking in with Local Networks

Although the identification and availability of local networks was viewed as a possible barrier in some locations, there were also positive and beneficial interactions that had taken place in others. Some academics had successfully established contact with local LGBTQ+ groups and organisations, finding them to be very supportive, whereas others had enlisted the help of willing gay individuals, such as friends and family members, to meet with students and share their experiences.

“I decided that I needed to review what we were doing but I needed help with it, so I approached the LGBTQ forum in [local city] and had a really helpful meeting with them. They agreed to come and do some teaching for us… I have learned how open and helpful LGBTQ people are when you talk to them. They are very kind, and they don’t mind, you know, they are very open, and they don’t take offence when you make a blunder. That’s been nice and confidence building for me.”(Blake)

“We have a lady who was born a man and she has been quite vocal about the needs of the trans gender community, and she has been doing some sessions talking about language and inclusivity.”(Charlie)

#### 4.5.3. LGBTQ+ Champions and Success

The support from others was appreciated by the academics and the appointment of an “LGBT Champion” was a novel approach in one university. The contribution made by students, especially those from the LGBTQ+ community, was also valued. It was recognised that their lived experiences of psychosocial issues were an asset to both the academics and the other students by providing a rich learning opportunity.

“They have just appointed a LGBT Champion for their programme development.”(Sam)

“We have had some students in our group that if it comes up, they will contribute to the conversation in class because I think students are much more open now than what they used to be. They are quite happy to talk about it and we can all learn as a result”(Jordon)

Academics who had successfully included and embedded LGBTQ+ health and addressed psychosocial issues in their nursing and midwifery pre-registration programmes willingly shared their learning experiences of what had worked for them.

“Be open to the unlearning. Be open to the fact that wherever your position is, whatever you are, whatever your personal identities, you don’t know everybody’s lived experiences or different group’s experiences. Be open to hear it.”(Ryan)

“I always refer the students back to the Nursing and Midwifery Council’s Code for Professional Conduct. The very first section in there is on prioritising people. I get them to look at reading every sentence one by one and trying to view it through the lens of sexualities and gender orientations… Just normalise it, throw it in. If you are doing a course on whatever you are doing it on, make sure you get some case studies in there, different ones, lesbian people, gay people, trans people.”(Joe)

## 5. Discussion

There is growing interest and attention regarding the scope and extent of the health inequalities and psychosocial needs of LGBTQ+ people [9]. Despite the evolving evidence, there appears to be a limited focus and attention within the education of health professionals, including nurses and midwives. There is some evidence of the inclusion of the needs of LGBTQ+ individuals, their families and communities within nursing and midwifery pre-registration programmes in the UK and Ireland. However, the design and delivery of LGBTQ+ specific learning materials remains patchy [32]. Although the current delivery is positive and welcome, there are barriers that need to be overcome to help ensure that LGBTQ+ health is developed as an integral and core component of all nursing and midwifery pre-registration programmes. There is an absence of a core curriculum upon which academics can draw, thereby ensuring that there is systematic delivery within programmes [33]. An important issue identified in the current study is the apparent lack of confidence on behalf of some academics of the most appropriate use of language, through concerns of “not wishing to cause offence”. Additionally, some academics described lacking knowledge, skills and confidence in the delivery of LGBTQ+ content in their programme, while attempting to “do their best” [34]. A further barrier identified that needs to be addressed is knowledge of and access to local LGBTQ+ networks [35]. It was recognised that these networks offer valuable sources of information and support, with LGBTQ+ workers and network members prepared to contribute to education sessions for students. The academics who participated in the current study were able to identify areas in their nursing and midwifery programmes where LGBTQ+ health could be incorporated in lectures, seminars and workshops, and skills simulation using case scenarios [36]. Consequently, there was recognition from study participants that more can and needs to be done to more fully integrate and reflect the needs of LGBTQ+ people. Therefore, arising from the study findings, are issues related to policy, education, practice learning and future research.

### 5.1. Policy

Despite the movement towards equality and diversity in healthcare, an inconsistency persists in the experiences of LGBTQ+ individuals in receiving appropriate and culturally competent healthcare [37]. Although the United Nations position on LGBTQ+ fundamental rights highlights the need to have in place LGBTQ+ specific policies for healthcare, it remains the case that not all healthcare authorities have embraced these recommendations [2,3]. However, LGBTQ+ people remain over-represented in healthcare and have specific health and psychosocial concerns that need to be identified and addressed [6,13]. Therefore, as a start, policies such as those espoused by the United Nations need to be embedded into educational curricula for healthcare professionals. This paper documented the limited focus and attention of specific healthcare and psychosocial needs of LGBTQ+ individuals within the education of health professionals, including nurses and midwives. Although some examples of good practice exist, most often this is dependent upon local leaders who have the capacity to champion LGBTQ+ specific issues in curricula, which results in an inconsistency in the design and delivery of appropriate material. To develop student cultural competence, it is therefore necessary to embed LGBTQ+ healthcare issues into the wider curricula and ensure that this addition has professional regulatory body backing. This strategic level of planning and commitment is necessary to ensure that LGBTQ+ health is fully and appropriately integrated within the standards for pre-registration programmes across all approved nursing programmes. In keeping with best practice, the design and delivery of material should be developed in collaboration with practice partners and local LGBTQ+ organisations. There is an onus on education providers to have appropriate policies in place to ensure that students are competent in identifying and disseminating best practice so that they can respond to and further minimise the already poor health outcomes experienced by LGBTQ+ people.

### 5.2. Education

Findings from the current study may be used to inform the direction of an LGBTQ+ evidence-based curriculum for future delivery within Schools of Nursing and Midwifery in the UK and Ireland, and more widely. Reports from participants in this study have generated several points that should be considered for inclusion in the curriculum. For example, Charlie stated that current language used makes assumptions about people’s sexuality, with steps towards rectification including introduction of “gender neutral” and “gender inclusive” language. This first fundamental point recognises that pronouns in the English language currently gender people as “she” or “he”, which may incorrectly label and possibly offend some non-binary people, and acknowledges that the term “non-binary” is an umbrella term that describes people who identify with a gender outside of classical gender pronouns, such as identifying as “female” or “male”. At present, this English language gender binary system assumes that all people classify as one of two genders, either female or male, when a non-binary person may consider themself to be:Agender—having no genderAndrogyne—somewhere in-between man or womanBigender—two gender identities at the same time or interchangeablyDemigirl/demiboy—not completely identifying as woman or man.

Since 2008, studies have reported LGBTQ+ people’s perceptions of microaggression, leaving some feeling marginalised by incorrect gender appointment, which has been shown to nurture negative outcomes, such as depression and low self-esteem [38]. Hence, it is important that nurses and midwives use individualised terminology, having ascertained the pronoun people would like to be addressed by and recording this within their health documentation. This concept of personalised pronouns is reinforced by Blake’s comment, which emphasises that gender identities should never be assumed.

Findings of this study also uncovered that no specific content with an LGBTQ+ exclusive focus currently exists to educate nurses and midwives about LGBTQ+ needs and concerns. This observation was also reported in a previous study [25]. As such, crucial learning resources are required to be developed for the purpose of assisting nurses and midwives to deliver modern contextualised LGBTQ+ care to service users. As part of this process, Schools of Nursing and Midwifery need to write LGBTQ+ focused educational learning aims and objectives, to then be mapped across programmes and the professional regulator’s standards and proficiencies, such as the Nursing and Midwifery Council Education Standards [39]. Standalone LGBTQ+ modules are also required to educate both pre-registration and post-registration nurses and midwives. As Chris narrated, embedding LGBTQ+ education into curricula has the added benefit of positively embracing those nursing and midwifery students who identify as being LGBTQ+. Among these developments, and as part of ongoing monitoring, public involvement is also necessary. Education delivery must be culturally focused and take a Person-Centred Care (PCC) approach, which recognises that service users’ individual resources, interests, needs and preferences are recognised and responded to [26,40,41]. As such, Ekman et al. [42] illustrated how person-centredness could be operationalised through PCC, with the theoretical framework encompassing the philosophy of personhood manifested through the patient narrative, partnership and coherent documentation, which are named as the three cornerstones of PCC.

### 5.3. Practice Learning 

Providing education that empowers nurses and midwives to provide culturally competent and effective care to trans and non-binary service users can be rehearsed within scenarios hosted within Schools of Nursing and Midwifery clinical skills labs. During the delivery process, the education delivered should take into account LGBTQ+ experiences of dissonance between their physical appearance, and their personal sense of being a man, woman, both or neither [43], with language used to refer to sexualised parts of the body named with words associated with individualised personal identity. In addition, nurses and midwives are required to acknowledge that gender identity is not necessarily fixed and may be fluid over time. To aid these processes, in 2020 the Royal College of Nursing recommended that nurses and midwives should:Be proactive in their approach to welcoming trans and non-binary service users to care.Always treat trans and non-binary patients in a respectful way, as they would other service users.When unsure about how to address a service user, begin by introducing themself with their own name and pronouns. Following on, the nurse or midwife can then politely and discreetly ask the person for their name and pronouns.Avoid disclosing a service user’s trans or non-binary status to anyone who does not need to know.Discuss issues relating to service user’s gender identity in private, and with care and sensitivity, such as how would they like parts of their body referred to and with appropriate language.

To ensure that important points are addressed within LGBTQ+ evidence-based curricula, an educational guide is required to be written. In addition, and as Ray highlighted, to aid delivery of an LGBTQ+ evidence-based curriculum, it is important that an “LGBTQ+ champion” is appointed who is not simply a volunteer. Within Schools of Nursing and Midwifery, this LGBTQ+ champion needs to be specifically prepared to undertake the role and allocated time within their workload to undertake the required activities. The designated function of the LGBTQ+ champion should be supported by a school education protocol that clearly outlines role aims, objectives, lesson plans, and methods of teaching and assessing students. A learning resource containing “sensitive” practice-based examples is required to accompany the LGBTQ+ curriculum and could, for example, be made available online for academics and students. 

Role-play scenarios can be used as learning activities to rehearse practice examples, thus providing a safe environment to explore situations where communication goes well or astray. During this process, an LGBTQ+ volunteer could be asked to participate in processes of development, validation and delivery. The benefits of practising scenarios are that students who learn within a safe environment are more likely to become safe, predictable and proficient practitioners. Moreover, due to the sometimes-sensitive nature of LGBTQ+ education, students may have conflicting experiences as they move from the classroom setting into clinical practice in relation to understanding user’s experiences and practice skills they have learned [44]. When such issues arise, academics can provide support, with referrals to the LGBTQ+ champion and their trained network when complexities arise. To draw together some of the key issues arising from the study, an example of an evidence-based curriculum is set out in Table 2.

## 6. Strengths and Limitations 

The study builds on the evolving and growing evidence base regarding the actions necessary to ensure that health professionals, including nurses and midwives, are prepared with the necessary knowledge and skills to effectively meet the needs of LGBTQ+ people. This is important because the UK and Ireland have seen significant changes and developments in legislation and policies aimed at improving the human rights and social inclusion of LGBTQ+ people within society. The findings from this study reflect the views and experiences of a sample of academics involved in the delivery of nursing and midwifery pre-registration programmes, and contribute to the identification of current education provision and areas where further developments are required. A strength of the current study is the inclusion of universities delivering nursing and midwifery pre-registration education across the UK and Ireland and the range of strategies currently in place. The researchers sought to be rigorous in all aspects of the research and acknowledge that not all Schools of Nursing and Midwifery participated in the study, despite best efforts. It is therefore possible that there is education activity taking place across the UK and Ireland that have not been reflected. These limitations are recognised and acknowledged.

## 7. Future Research

Improving healthcare professionals’ competency is a crucial part of interventions to reduce health inequalities. Specifically, the promotion of education regarding LGBTQ+ health issues and concerns across educational institutions can empower healthcare professionals, leading ultimately to a “cascade effect” whereby the LGBTQ+ patient benefits from a competent and culturally sensitive workforce at a time they are most vulnerable. To ensure that changes to curricula are having the desired impact in practice and to identify other areas of need, future research should address both LGBTQ+ experiences of healthcare and service providers’ experiences of delivery. This research should be multi-centre and longitudinal in design to reflect changes in educational curricula and the impact and outcomes achieved. Although the implementation of education will go some way to address the inequalities experienced by LGBTQ+ people receiving healthcare, further research is needed to ensure that changes to curricula encompass appropriate and up-to-date LGBTQ+ inclusive care. The ensuing impact on practice and the development of positive attitudes and values towards LGBTQ+ people should also be explored. Finally, LGBTQ+ is an umbrella term that has limitations, particularly in the context of appropriate and sensitively delivered healthcare. Future research that addresses the specific needs of the sub-populations of this group should be integrated into educational curricula so that it does not undermine individual care needs. This is important for future research, because each sub-group has different and distinct needs that should be reflected in health and educational policies. Although an opportunity exists for LGBTQ+ learning for students in healthcare sectors, specifically in nursing and midwifery, it is essential that this learning is facilitated in a way that is truly inclusive and equally accessible to all. 

## 8. Conclusions

Nurses and midwives are central to the effective delivery of culturally competent and person-centred healthcare by responding to the needs of individuals, families and communities. LGBTQ+ people have distinct health needs, yet often these needs remain unidentified and unmet, thereby contributing to their on-going health inequalities, poor health outcomes and marginalisation. With education and development, nurses and midwives have the potential to make significant contributions to improving the care experiences and health outcomes of LGBTQ+ people. For the potential of these health professionals to be fully realised, they must be prepared with the necessary knowledge, skills and attitudes. However, current provision remains inconsistent and is in need of sustained and ongoing development. Action is required by all Schools of Nursing and Midwifery to provide pre-registration nursing and midwifery education to ensure that the future nursing and midwifery workforce is adequately prepared. 

## Figures and Tables

**Figure 1 ijerph-18-11366-f001:**
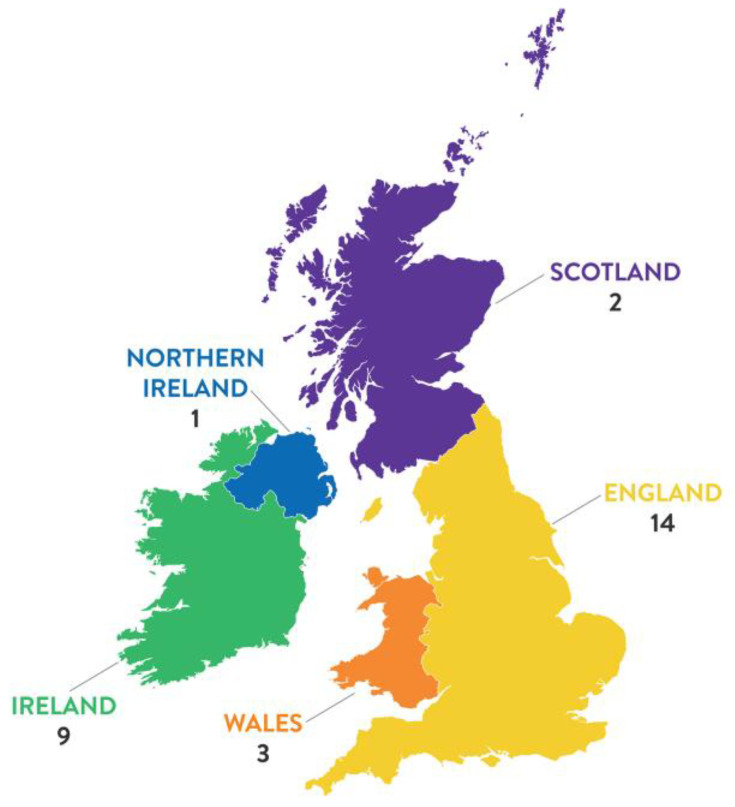
Distribution of survey respondents across the UK and Ireland.

**Table 1 ijerph-18-11366-t001:** Participant information by programme.

Programme	Survey Responses	Interviews	Participant Pseudonym				
Nursing	14	5	Alex	Chris	Finn	Ryan	Taylor
Midwifery	9	4	Blake	Charlie	Jordon	Ray	
Nursing and midwifery	6	3	Joe	Lee	Sam		
Total	29	12					

**Table 2 ijerph-18-11366-t002:** Summary of potential content of evidence-based LGBTQ+ education for nurses and midwives.

No	Potential Content
1	Develop evidence-based LGBTQ+ educational guidelines to assist nurses and midwives to deliver modern contextualised LGBTQ+ care to service users.
2	Devise standalone module descriptors with an LGBTQ+ focus, which are designed to empower nurses and midwives to provide effective care to trans and non-binary service users and which include the points herein (3, 4, 5, 6, 7, 8, 9, 10, 11, 12, 13, 14, 15, 16, 17) and also learning objectives cited in McCann et al. (2021) (Outline aims and objectives for lesson plans, and methods of teaching and assessing students’ skills in delivery).
3	Include a guide of relevant “non-binary” definitions, e.g., agender, androgyne, bigender, demigirl and demiboy, and provide contextual examples from nursing and midwifery practice.
4	Include a guide of “gender neutral” and “gender inclusive” language for nurses and midwives.
5	Provide examples of appropriate LGBTQ+ language to use when addressing and discussing care provision with service users.
6	Underpin LGBTQ+ education with an evidence-based Person-Centred Care (PCC) approach, which considers individual’s resources, interests, needs, and preferences, which includes: (1) care users’ narratives, (2) partnership in care, and (3) coherent documentation.
7	Ensure that all objectives are underpinned by an evidence base and use relevant research papers to underpin educational points made.
8	Involve members of the public in educational developments of LGBTQ+ education, ensuring that LGBTQ+ care users, staff, and students see themselves reflected in curriculum.
9	Develop sensitive scenarios and toolkits for rehearsal and role play within the classroom, on-line and/or clinical skills labs, which explore examples of where LGBTQ+ relevant communication can go well or astray.
10	Critically appraise the term microaggression and provide contextual examples of good and bad practice.
11	Discuss methods of documenting service users preferred terminology, pronouns, and language used to refer to parts of their body.
12	Outline legal requirements for confidentiality and avoiding disclosure of service user’s trans or non-binary status to unnecessary people.
13	Discuss social identity confusion which can occur when some LGBTQ+ people experience dissonance between their physical appearance and personal sense of being a man, woman, both, or neither and how gender identity is not necessarily fixed but fluid over time (emphasise importance of not making assumptions about people’s sexuality).
14	Outline methods of embedding LGBTQ+ teaching and learning across nursing and midwifery curriculum.
15	Plan systems to support students who have conflicting experiences, as they move from the classroom out into clinical practice.
16	Emphasise the importance of appointing a designated “LGBTQ+ champion” and outline their role and training required to effectively undertake this position.
17	Gain support from professional bodies to endorse teaching aids developed.

## Data Availability

The data presented in this study are available on request from the corresponding author. The data are not publicly available due to restrictions included in the informed consent signed by the study participants.

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
