# Peer review of "LGBTQ+ Psychosocial Concerns in Nursing and Midwifery Education Programmes: Qualitative Findings from a Mixed-Methods Study"

_ijerph, 2021, doi:10.3390/ijerph182111366_

Round 1

Reviewer 1 Report

An interesting and important topic.The title should changed, however, as this is a report of the qualitative findings from a mixed methods study and not a qualitative study as stated.

I believe the first paragraph of the introduction is probably not necessary, and the paper would be better off starting directly with discussion of LGBQT+ health. The introduction generally is somewhat redundant with statements about the importance of health care workers and the need for meeting the needs of different populations.

More information is needed about the qualitative methodology, as it appears to be thematic analysis but this is not stated or justified.

p2. line 49--subject verb agreement errors

p2--not sure what you refer to by "many LGBTQ+ people have 52 specific cultural and psychosocial needs" as this is true of all people. Please clarify. The last sentence of this paragraph is also too long and does not really make sense--I am not sure people can demonstrate their values, nor that any specific ones are necessary to the care of LGBTQ+ persons. All providers should provide care without judgement or need to justify choices based on their values.

p3. 29 responses seems a very small sample size if all schools were queried. Did you request only one response per school? If so, from whom and why?

Author Response

Reviewer 1

The title should change, however, as this is a report of the qualitative findings from a mixed methods study and not a qualitative study as stated.

The title has been changed to reflect the study design.

Introductory paragraph

The introductory paragraph has been edited and rewritten for clarity.

More information is needed about the qualitative methodology, as it appears to be thematic analysis but this is not stated or justified

The methodology section has been edited and updated and includes thematic analysis.

P2. line 49--subject verb agreement errors

Sentence revised

P2.

Paragraphs and sentences reviewed and edited throughout page 2

p3. 29 responses seem a very small sample size if all schools were queried. Did you request only one response per school? If so, from whom and why?

Sampling and participants clarified on page 3

Reviewer 2 Report

- Contextualize the study within the interests, aims, and models of the transcultural nursing discipline.

- Add further data to contextualize the political-geographical, academic, and sociocultural environment. What reasons might be there to set the investigation in Ireland/UK (other than the authors' employment in those countries)?

- The designation of the mixed design needs to be reconsidered. No meaningful information is provided on the quantitative data derived from the questionnaire.

- Other data on the participants interviewed (such as whether or not they belonged to the group studied, their age, and their contractual links with the universities for which they work) would constitute an improvement in the rigor and quality criteria of the study.

- The analysis of the data needs further clarification.

Author Response

Reviewer 2

Contextualize the study within the interests, aims, and models of the transcultural nursing discipline

Revised and updated within text

Add further data to contextualize the political-geographical, academic, and sociocultural environment. What reasons might be there to set the investigation in Ireland/UK (other than the authors' employment in those countries)?

Revised and updated within text

The designation of the mixed design needs to be reconsidered. No meaningful information is provided on the quantitative data derived from the questionnaire

Clarification of the extraction of and presentation of the qualitative data has been included for clarity

Other data on the participants interviewed (such as whether or not they belonged to the group studied, their age, and their contractual links with the universities for which they work) would constitute an improvement in the rigor and quality criteria of the study.

Details of the participant details that were collected have been presented

The analysis of the data needs further clarification.

The methods section has been reviewed and updated throughout